# Myeloid - derived suppressor cells in Type 1 diabetes are an expanded population exhibiting diverse T-cell suppressor mechanisms

**Anna Grohová**[1]*, **Klára Dáňová**[1,2], **Irena Adkins**[1,2], **Zdeněk Šumník**[3],
**Lenka Petruželková**[3], **Barbora Obermannová**[3], **Stanislava Koloušková**[3], **Radek Špíšek**[1,2],
**Lenka Palová-Jelínková**[1,2]

1 Department of Immunology, Second Faculty of Medicine, Charles University and Motol University Hospital, Prague, Czech Republic, 2 SOTIO a.s., Prague, Czech Republic, 3 Department of Pediatrics, Second Faculty of Medicine, Charles University and Motol University Hospital, Prague, Czech Republic

* anna.grohova@gmail.com

**Data Availability Statement:** All relevant data are within the paper and its Supporting Information files.

## Abstract

Myeloid-derived suppressor cells (MDSC) represent a heterogeneous group of immature myeloid cells with immunoregulatory function in cancer and autoimmune diseases. In humans, two subsets of MDSC were determined based on the characteristic surface markers, monocytic MDSC (M-MDSC) and granulocytic MDSC (G-MDSC). Expansion of MDSC has been reported in some murine models and patients with autoimmune diseases and their immune-suppressive properties were characterized. However, the exact role of MDSC in the pathogenesis of autoimmune diseases is more complex and/or controversial. In type 1 diabetes mellitus (T1D), the increased frequency of MDSC was found in the blood of T1D patients but their suppressor capacity was diminished. In our study, we assessed the role of M-MDSC in the pathogenesis of T1D and showed for the first time the increased frequency of M-MDSC not only in the blood of T1D patients but also in their at-risk relatives compared to healthy donors. T1D patients with inadequate long term metabolic control showed an elevation of M-MDSC compared to patients with better disease control. Furthermore, we described the positive correlation between the percentage of M-MDSC and Th17 cells and IFN-γ producing T cells in T1D patients and their at-risk relatives. Finally, we found that the ability of M-MDSC to suppress autologous T cells is efficient only at the high MDSC: T cells ratio and dependent on cell-cell-contact and TGF-β production. Our data show that the engagement of MDSC in the pathogenesis of T1D is evident, yet not entirely explored and more experiments are required to clarify whether MDSC are beneficial or harmful in T1D.

## Introduction

Type 1 diabetes (T1D) is a chronic metabolic disorder resulting from the break of self-tolerance followed by the expansion of auto-reactive T cells leading to immune-mediated

**Funding:** Lenka Palová Jelinková, Klára Dáňová, Irena Adkins, and Radek Spisek are employees of Sotio a.s., Prague, Czech Republic. The funder provided support in the form of salaries for authors (LPJ, KD, IA, RS) but did not have any additional role in the study design, data collection and analysis, decision to publish, or preparation of the manuscript. The specific roles of these authors are articulated in the 'author contributions' section. The other co-authors declare that the research was conducted in the absence of any commercial or financial relationship that could be construed as a potential conflict of interest. This study was supported by the project of the Ministry of Health, Czech Republic, for conceptual development of the research institution 00064203 (University Hospital Motol, Prague, Czech Republic).

**Competing interests:** Lenka Palová Jelinková, Klára Dáňová, Irena Adkins, and Radek Spisek are employees of Sotio a.s., Prague, Czech Republic. Lenka Palová-Jelínková, Klára Dáňová, and Radek Špíšek are named inventors in the patent, "Tolerogenic Dendritic Cells, Methods of Producing the Same, and Uses Thereof" (U.S. Provisional Application No. 62/066,994), which describes methods for the preparation of stable semi-mature tolerogenic DC. This does not alter our adherence to PLOS ONE policies on sharing data and materials. The other authors have no financial conflicts of interest.

destruction of β-cells in the pancreas and long-term hyperglycemia [1, 2]. Although the previous knowledge assumed the genetic predisposition of T1D [3–5], currently, the screening program for T1D is also based on the presence of specific autoantibodies. Thus the presence of these specific autoantibodies could identify the potential risk of diabetes in close relatives of patients with T1D [6]. Nowadays, there is no casual treatment either for diabetic patients or their at-risk relatives. However, much effort has been done to develop immune intervention strategies based on the cell therapy to treat and prevent T1D, especially in the subjects at risk with preserved β-cells function.

Myeloid-derived suppressor cells (MDSC) represent a heterogeneous group of immature myeloid cells with the immune suppressive activity which makes them attractive targets for the treatment of autoimmune diseases. However, the controversial role of MDSC in various autoimmune diseases was observed. The number of MDSC rises during pathological conditions, including cancer, autoimmune conditions, inflammation, trauma, parasitic infections, sepsis, and transplantation [7–12]. In mice, MDSC are characterized by the expression of cell surface markers CD11b and Gr-1 and two functional subsets have been described, $Ly6G^+/Ly6C^{low}$ referred to as granulocytic MDSC (G-MDSC) and $Ly6G^-/Ly6C^{high}$ referred as monocytic MDSC (M-MDSC) [13, 14].

In humans, MDSC lack the characteristic markers and have been identified as $HLADR^{low/neg}$ $CD11b^+CD14^+CD15^-CD33^{high}$ monocytic MDSC (M-MDSC) and $HLADR^{low/neg}$ $CD11b^+$ $CD14^-CD15^+CD33^{low/mid}$ granulocytic MDSC (G-MDSC) [15–17]. Both M-MDSC and G-MDSC are immune suppressive but utilize different pathways to regulate T cell, B cell, and NK cell functions. In M-MDSC a dominant mechanism has been attributed to arginase-1, iNOS, and ROS production which together leads to T cells inhibition and apoptosis and expansion of T regulatory cells (T regs) [18–21]. The inhibitory effect is also cell-cell contact-dependent [8]. The role of MDSC in regulating B cell is less understood despite several studies in humans and murine models confirming the interplay between MDSC and B cells [22–24]. However, more data is needed for further characterization of particular subsets of human MDSC and their precious function.

MDSC have been studied mostly in cancer patients, where MDSC inhibit the function of effector cells and antigen-presenting cells in the tumor microenvironment through various mechanisms such as the deprivation of arginine and cysteine, production of NO and ROS, IL-10 and TGF-β, or by an expansion of Tregs [25, 26]. However, the exact role of MDSC in the pathogenesis of autoimmune diseases is more complex and controversial revealing completely opposite characteristics from rather pro-inflammatory to disease protective in various autoimmune conditions. A potentially disease-protective effect of MDSC was reported in a murine model of multiple sclerosis (MS), where MDSC inhibited autoreactive Th1 and Th17 responses [27] or in rheumatoid arthritis (RA) where MDSC reciprocally regulate Th17/T regs via IL-10 production [28]. Next, the protective role of MDSC was documented in the inflammatory bowel disease (IBD) and the skin with contact eczema [29, 30]. In contrast, MDSC were reported to promote Th17 differentiation with subsequent increase in the disease severity in systemic lupus erythematosus (SLE) but also in RA [31–34]. Regarding T1D, MDSC prevented diabetes onset in the adoptive co-transfer in the NOD-SCID model [35]. Nevertheless, limited data is available about the role and suppressive mechanism of MDSC in patients with active T1D or subjects who are at risk of diabetes onset. Whitfield-Larry et al. observed an increased frequency of MDSC in the peripheral blood of T1D patients but these native MDSC were not maximally suppressive compared to cytokine-induced MDSC that were generated *in vitro* from patient´s PBMC [36].

Our study aimed to assess the frequency and function of MDSC not only in T1D patients but also in their at-risk relatives and to explore the association of MDSC with Th17 cells and

IFN-γ producing T cells. We showed for the first time the MDSC expansion in T1D patients and also in their at-risk relatives. Next, we analyzed if there is any relationship between the frequency of MDSC and the level of glycated hemoglobin as a marker of the metabolic condition of T1D patients reflecting the long-term glycemia level. Further, we explored in detail the molecular mechanisms of suppression in MDSC directly isolated from the peripheral blood of T1D patients and observed the necessity of the cell-cell contact between MDSC and T cells and the importance of TGF-β production in MDSC to fulfill their immunosuppressive potential.

Based on the present work we postulated that the engagement of MDSC in the pathogenesis of T1D is indisputable, yet not fully clarified and more experiments are required to clarify the precise role of M-MDSC in T1D pathogenesis.

## Materials and methods

### Subjects

Blood samples were collected from 65 patients diagnosed with T1D and from 21 their first degree relatives with positive islet-specific autoantibodies (anti-GAD, anti-IAA and anti IA-2), considered as at-risk relatives, and from 24 healthy donors (HD) in corresponding age. Subject's demographics are summarized in Tables 1 and 2. Further 4 adult patients with the diagnosis of lung cancer (squamous cell lung carcinoma) were included. Patients were selected as pediatric patients up to the age of 18 years with both recent onset or long-term T1D. The blood collection of patients with a recent T1D onset was performed after the metabolic stabilization and after the establishment of normoglycaemia. At the time of the blood collection, none of the T1D patients had diabetic ketoacidosis, nor any active infection and other comorbidities, except long-term controlled comorbidities associated with T1D (thyroiditis, celiac disease). The first-degree relatives were subjects up to the age of 18 years whose at least one sibling suffer from T1D manifested up to the age of 20 years. These subjects were analyzed for HLA DQB1, DQA1 genotyping, and tested for islet-specific autoantibodies. The risk of T1D was assessed based on the HLA genetic association study in Czech children and the positivity of at least one of the tested autoantibodies [37].

Patients with T1D were divided into two groups: Group A containing the subjects with high glycated hemoglobin (HbA1c) and Group B containing the subjects with lower glycated hemoglobin. HbA1c $\leq$ 7.5% (58 mmol/mol) was the cut-off level according to American Diabetes Association [38]. All subjects provided written consent and the study was approved by the local Ethics Committee for Multi-Centric Clinical Trials of the University Hospital Motol in Prague, Czech Republic on 26th July 2015.

### Isolation of CD14$^+$ HLA-DR$^{low/neg}$ CD33$^+$ MDSC by sorting

Peripheral blood mononuclear cells (PBMC) were isolated from fresh peripheral blood by using density centrifugation (Ficoll-Paque; GE Healthcare). PBMC were resuspended in cold MACS buffer (Miltenyi Biotec) and incubated with CD33 MicroBeads (Miltenyi Biotec) for 15

**Table 1. Characterization of subjects evaluated in the study.**

|  | T1D | subjects at-risk | HD |
|---|---|---|---|
| **Number (female/male)** | 65 (36/29) | 21 (11/10) | 24 (8/16) |
| **Age: mean ± SD; range (y)** | 14,6 ± 2,3; 6–18 | 9,7 ± 2,8; 4–18 | 16,2 ± 2,9; 10–20 |
| **Ethnicity** | Czech (of European descent) | Czech (of European descent) | Czech (of European descent) |

T1D, Type 1 diabetes; HD, healthy donors.

**Table 2. Laboratory parameters of T1D subjects.**

| T1D subjects | |
|---|---|
| T1D duration: mean ± SD; range (y) | 4,5 ± 2,9; 0–12 |
| HbA1c: median; range (mmol/mol) | 57; 45–120 |
| TAG: median; range (mmol/l) | 1,35; 0,46–1,9 |
| Total cholesterol: median; range (mmol/l) | 4,45; 3,3–6,1 |
| HLD–cholesterol: median; range (mmol/l) | 1,17; 0,71–1,59 |
| C-peptide: median; range (pmol/l) | 3,3; 3,3–796 |
| % of subjects with positive GAD-AB | 63,1 |
| % of subjects with positive ANTI-IAA | 55,2 |
| % of subjects with positive IA2-AB | 78,9 |

T1D, Type 1 diabetes; HbA1c, glycated hemoglobin; TAG, triglycerides, GAD-AB, anti-glutamic acid decarboxylase antibody; ANTI-IAA, anti-insulin antibody; IA2-AB, anti-tyrosine phosphatase antibody.

min on ice. Then cells were washed with cold MACS buffer to remove unbound beads and subsequently subjected to depletion of CD33 negative cells on MACS column® (Miltenyi Biotec) according to the manufacturer´s instructions. The CD33 positive cell fraction was collected, washed and stained with anti-CD14 (Pacific Blue™ anti-human CD14 Ab, Biolegend) and anti-HLA-DR (Alexa Fluor® 700 anti-human HLA-DR Ab, Biolegend) for 20 min at 4˚C. Cells were then washed with PBS and the $CD14^+$ HLA-DR$^{low/neg}$ $CD33^+$ MDSC were then sorted using S3e cell sorter (Biorad).

## Flow cytometry and gating strategy

PBMC were isolated from fresh peripheral blood by using density centrifugation (Ficoll-Paque; GE Healthcare). For the analysis of MDSC, PBMC were surface labeled with anti-CD14-BD Horizon V450 (BD Biosciences), anti-HLA-DR-Alexa Flour 700 (Biolegend), anti-CD11b-FITC (eBioscience), anti-CD33-PE-Cy7 (Biolegend), anti-CD15-APC (eBioscience). The gating strategy is depicted in S1 Fig. Expression of CD3ζ was analyzed on PBMC by intracellular staining using a fixation/permeabilization buffer kit (eBioscience) with appropriate mAb anti-CD3ζ-PE (Exbio). For the analysis of IFN-γ and IL-17A producing cells, PBMC were stimulated with PMA (50 ng/ml; Sigma-Aldrich) plus ionomycin (1 mg/ml; Sigma-Aldrich) for 4h in the presence of brefeldin A (5 mg/ml; Biolegend). Intracellular IFN-γ and IL-17A staining were assessed using a fixation/permeabilization buffer kit (eBioscience) with appropriate mAb IFN-γ-Pacific Blue (Biolegend) and anti-IL-17-A647 (Biolegend) and measured by flow cytometry. Data were acquired by LSRFortessa and LSR II flow cytometers (BD Biosciences) and analyzed using FlowJo software (Tree Star).

## Immunosuppression assays with MDSC and cytokine detection

Autologous T cells, selected as a whole $CD3^+$ population from PBMC using EasySep™ Human T Cell Isolation Kit (StemCell Technologies), 5× $10^4$ cells per well were labeled with CFSE (1 nM) and activated by anti-CD3/CD28 expander beads ($2.5 × 10^5$ beads per well) (Thermofisher). 1 hour after activation with anti-CD3/CD28 beads, T cells were incubated with different ratios of sorted MDSC (1:1, 1:2, 1:4 MDSC/ T cell ratio). T cell proliferation was measured as CFSE dilution using flow cytometry on day 6. To assess the mechanism of MDSC-mediated T cell inhibition, the particular inhibitors were added to the MDSC-T cell culture in the determined dosage. The subsequent proliferation of T cells was analyzed by flow cytometry. We used L-NMMA–iNOS inhibitor (100μM, Sigma), anti-TGF-β mAb (20 μg/ml) (LEAF™

Purified anti-humanTGF-β Ab, Biolegend) and Nor-NOHA–arginase-1 inhibitor (300μM, CaymanChemical). Cytokine detection (IFN- γ, IL-17) in cell culture supernatants were evaluated by ELISA (R&D Systems).

## Transwell chamber assay

For Transwell assays, M-MDSCs were added to the Transwell inserts to separate from autologous or allogeneic T cells. T cells purified from blood PBMC as a whole CD3$^+$ population using EasySep™ Human T Cell Isolation Kit (StemCell Technologies) were labeled with CFSE (1nM) and activated by CD3/CD28 beads 1h prior assay, than added to the lower compartment of a plate containing Tranwell insert. M-MDSC, sorted from PBMC by S3e cell sorter (Biorad) were added to the upper compartment in an indicated ratio (1:1, 1:2 and 1:4 T cell/MDSC). They were separated from the lymphocytes by an 8μm pore. T cell proliferation was measured as CFSE dilution using flow cytometry on day 6. Transwell plates were purchased from EMD Millipore (Billerica).

## Statistical analysis

The data were tested with Shapiro-Wilk normality test and then the statistical analyses were performed with a two-tail paired/unpaired t-test or Mann-Whitney test using GraphPad Prism 6. Probability levels for correlation were determined using the Spearman correlation test. A p-value $\leq$ 0.05 was considered statistically significant. Due to limited amounts of blood samples, not all subjects are involved in all experiments.

## Results

### M-MDSC are expanded in the blood of T1D patients and their first degree relatives who are at risk of diabetes onset

To analyze whether the M-MDSC are expanded in subjects with the autoimmune condition we collected blood samples of T1D patients, their first-degree relatives with positive islet-specific autoantibodies, considered as at-risk relatives, and healthy donors (HD). The frequency of CD14$^+$ HLA-DR$^{low/neg}$ CD11b$^+$CD33$^{high}$ M-MDSC in PBMC freshly isolated from blood samples was determined by flow cytometry.

Compared to HD, patients with T1D as well as their at-risk relatives exhibited a noticeable expansion of CD14$^+$ HLA-DR$^{low/neg}$ CD11b$^+$CD33$^{high}$ M-MDSC in the peripheral blood (Fig 1A). Next, we evaluated whether the long-term hyperglycemia affected the expansion of M-MDSC since chronic hyperglycemia was referred to as a strong pro-inflammatory condition [39–43]. The T1D patients were divided into two groups based on the level of glycated hemoglobin HbA1c. The frequency of M-MDSC was significantly higher in Group A containing patients with HbA1c of > 7.5% compared to Group B containing patients with HbA1c of $\leq$ 7.5% (Fig 1B). However, correlation analysis did not show any statistically significant association between the frequency of M-MDSC and HbA1c levels (Fig 1C). Similarly, no significant correlation was observed between the M-MDSC number and the disease duration (Fig 1D).

### The strong positive correlation between the percentage of M-MDSC and percentage of Th17 cells and IFN-γ$^+$ T cells was found in T1D patients, at-risk relatives, but not in healthy donors

CD4$^+$IL-17$^+$ Th17 cells and CD4$^+$IFN-γ$^+$ T cells were documented to play an important role in the pathogenesis of T1D [44–46], however, the association between MDSC and

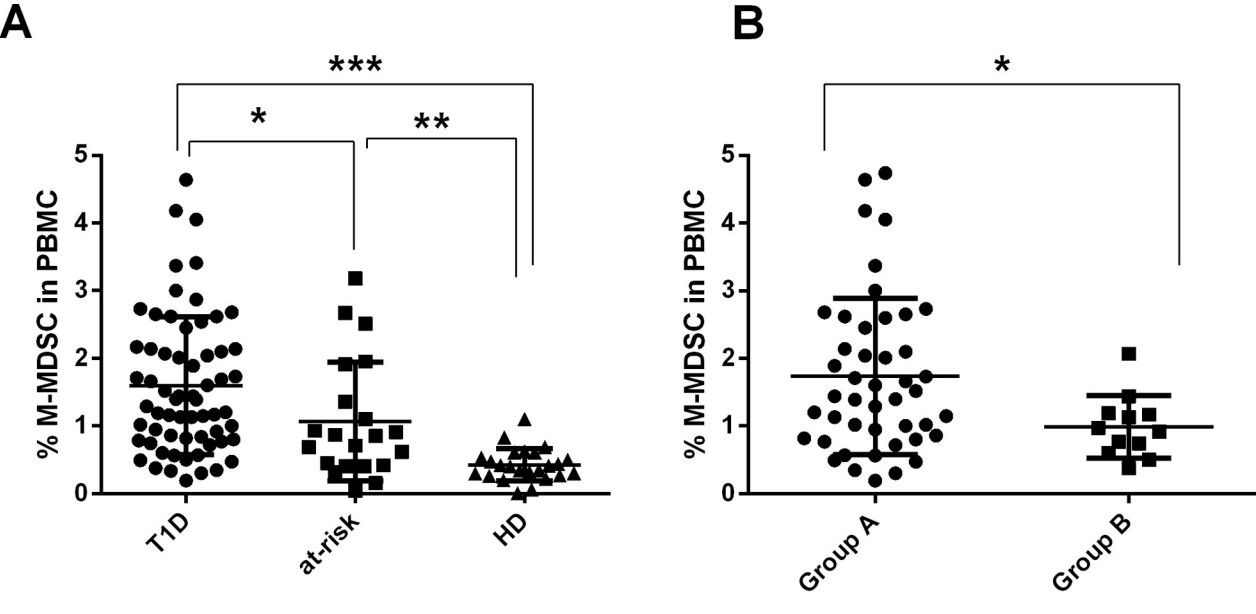

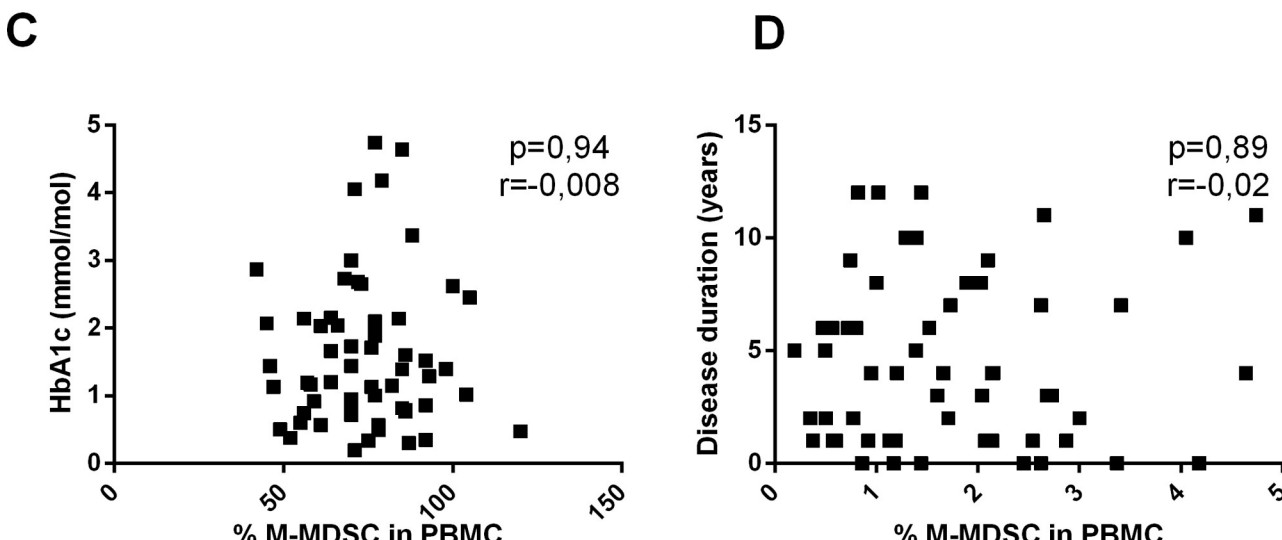

**Fig 1. The frequency of M-MDSC in the peripheral blood of T1D patients and their relatives and its relation to the long-term disease control and the disease duration.** (A) The frequency of M-MDSC in PBMC of T1D patients (T1D, n = 63), their first degree relatives (at risk, n = 21) and healthy donors (HD, n = 24) was determined by flow cytometry. Compared to HD, T1D patients, as well as their relatives at risk, exhibited a significant expansion of M-MDSC in the peripheral blood. *p≤0.05, **p≤0.01, ***p≤0.001 (Mann-Whitney test). (B) The T1D subjects were divided based on the level of HbA1c; Group A, HbA1c of > 7.5% (n = 52) and Group B, HbA1c of ≤ 7.5% (n = 12). The frequency of M-MDSC in PBMC was significantly higher in the Group A. *p≤0.05, **p≤0.01, ***p≤0.001 (Mann-Whitney test). (C) The correlation analysis between the HbA1c level and the frequency of M-MDSC in PBMC of T1D subjects (n = 58) was performed. No statistically significant correlation was found. Each point represents the value from an individual subject. The r value indicates the correlation index according to Spearman correlation analysis. (D) The correlation analysis was performed between the frequency M-MDSC in PBMC of T1D subjects (n = 57) and the time since the disease onset. No significant correlation was observed. Each point represents the value from an individual subject. The r value indicates the correlation index according to Spearman correlation analysis.

proinflammatory T cell populations has not been investigated in T1D yet. Thus, we examined the possible correlation between the frequency of pro-inflammatory T cell subpopulations such as CD4$^+$IL-17$^+$ Th17 cells and CD4$^+$ IFN-γ$^+$ T cells and M-MDSC in our subjects. First, we evaluated the levels of CD4$^+$IL-17$^+$ Th17 cells and CD4$^+$IFN-γ$^+$ T cells in freshly isolated PBMC from T1D, at-risk relatives and HD by flow cytometry. As shown in Fig 2A, the significant expansion of CD4$^+$IL-17$^+$ Th17 cells in PMBC of T1D patients and at-risk relatives was noted in comparison to HD, respectively. The frequency of CD4$^+$IFN-γ$^+$ T cells was increased in T1D subjects and at-risk relatives compared to HD although the increase was statistically significant only between HD and T1D patients. Next, we asked whether the proinflammatory T cells rely on M-MDSC as MDSC are considered to suppress immune response. The correlation analysis was performed between the frequency of M-MDSC and CD4$^+$IL-17$^+$ Th17 cells and CD4$^+$IFN-γ$^+$ T cells in PBMC. However, we observed a positive correlation between M-MDSC and the frequency of CD4$^+$IL-17$^+$ Th17 cells in T1D patients and at-risk relatives but not in healthy donors. Moreover, a strong positive correlation between the frequency of CD4$^+$IFN-γ$^+$ T cells and the frequency of M-MDSC was observed in T1D patients and their at-risk relatives. In healthy donors, the frequency of M-MDSC and CD4$^+$IFN-γ$^+$ T cells showed a negative correlation (Fig 2B and 2C).

Our data suggest the interplay between proinflammatory T cells and M-MDSC with a reciprocal expansion of MDSC under inflammatory conditions in T1D patients and their at-risk relatives. Despite this fact, M-MDSC keep their immune suppressive potential *in vitro* as discuss below.

## M-MDSC are T cell suppressors but only at high MDSC: T cell ratio

The previous study documented that cytokine-expanded CD33$^+$ MDSC from T1D patients and healthy donors equally suppressed allogeneic T cell proliferation, whereas CD33$^+$ MDSC purified from the blood of T1D patients have diminished suppressive function *in vitro* in terms of reducing the proliferation of T cells isolated from healthy donors [36].

In our study, we thought to determine the capacity of M-MDSC directly isolated from the fresh peripheral blood of T1D patients to suppress autologous as well as allogeneic T cell proliferation. For this purpose, M-MDSC sorted from PBMC were titrated into the cultures comprising of autologous or allogeneic T cells selected as a whole CD3$^+$ population and activated by anti-CD3/CD28 beads 1h prior to co-culturing with M-MDSC. Whereas M-MDSC from healthy donors exhibited only a marginal effect on autologous T cell proliferation, M-MDSC from T1D patients significantly inhibited autologous CD4$^+$ as well as CD8$^+$ T cell proliferation in a dose-dependent manner. The inhibition of T cell proliferation by M-MDSC was the most effective at the 1:1 ratio of MDSC: T cell, however, the maximal suppression was about 50% at MDSC: T cell ratio 1:1. The inhibitory function was lost at the 1:4 ratio (Fig 3A).

Likewise in the autologous system, the proliferation of CD4$^+$ as well as CD8$^+$ T cells was suppressed by allogeneic M-MDSC sorted from PBMC of T1D patients in a dose-dependent manner (Fig 3B).

Consistently with the fact that M-MDSC induced the inhibition of CD4$^+$ and CD8$^+$ T cell proliferation, the production of T cell proinflammatory cytokines IFN-γ and IL-17 was also diminished in the culture with M-MDSC as measured by ELISA. The significantly reduced concentration on both pro-inflammatory cytokines was observed in the cultures of activated autologous T cells from T1D with matching M-MDSC, isolated by sorting, at 1:1 ratio. As we expected we did not observe the significant suppression of T cell pro-inflammatory cytokines production in healthy donors (Fig 3C) likewise M-MDSC from HD did not effectively suppress the T cell proliferation.

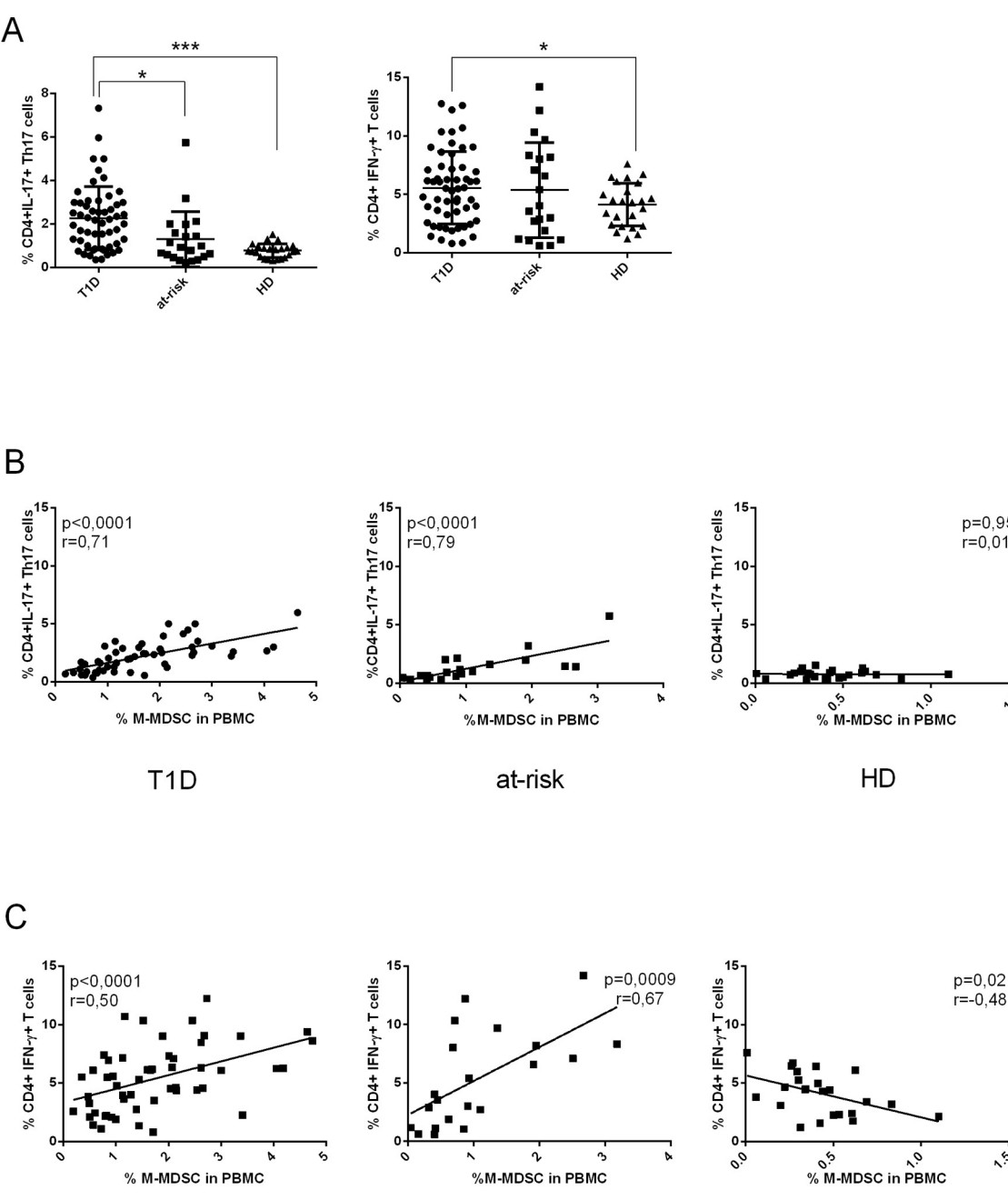

**Fig 2. The correlation between the percentage of M-MDSC and proinflammatory T cells.** (A) The frequency of CD4+IL-17+ Th17 cells and CD4+ IFN-γ+ T cells in PBMC was increased in T1D patients (n = 55) and their at-risk relatives (n = 21) compared to healthy donors (HD) (n = 24). *p≤0.05, **p≤0.01, ***p≤0.001 (Mann-Whitney test). (B) The frequency of CD4+IL-17+ Th17 cells was in a strong correlation with the frequency of M-MDSC from PBMC in T1D patients and at-risk relatives. No correlation was found between the frequency of M-MDSC and Th17 cells in HD. Each point represents the value from an individual subject. The r value indicates the correlation index according to Spearman correlation analysis. (C) The frequency of CD4+ IFN-γ+ T cells positively correlates with the frequency of M-MDSC from PBMC in T1D patients and their at-risk relatives. In contrast, a moderate negative correlation was found between the frequency of M-MDSCs and CD4+ IFN-γ+ T cells in HD. For correlation analyses, each point represents the value from an individual subject. The r values indicate the correlation index according to Spearman correlation analysis.

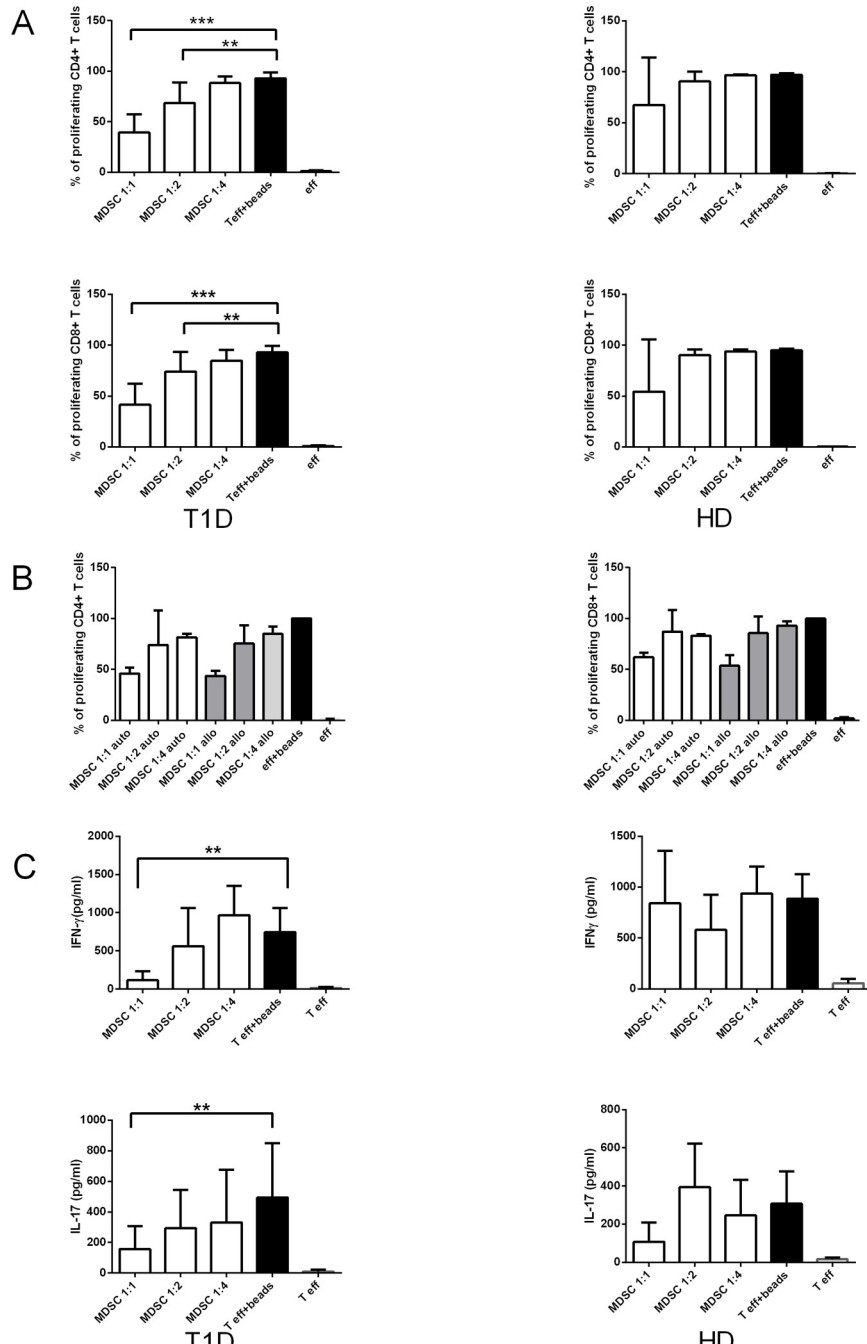

**Fig 3. M-MDSC from T1D patients suppress CD4⁺ and CD8⁺ T cell proliferation and T cell proinflammatory cytokines production.** (A) M-MDSC sorted from PBMC of T1D patients (n = 15), and healthy donors (HD) (n = 3) were co-cultured with autologous CD4⁺ and CD8⁺ T cells activated by antiCD3/CD28 beads. M-MDSC from T1D patients significantly inhibited T cell proliferation in a dose-dependent manner, the MDSC/ T cell ratio of 1:1 was the most effective, and the inhibitory function was lost at 1:4 ratio. M-MDSC from HD had only a slight effect on T cell proliferation in any MDSC/ T cell ratio. *p≤0.05, **p≤0.01, ***p≤0.001 (paired t-test). (B) Sorted M-MDSC from T1D patients (n = 2) were co-cultured in the different ratios (1:1, 1:2 and 1:4) with autologous and/or allogeneic T cells. M-MDSC suppressed equally proliferation of autologous as well as allogeneic CD4⁺ T cells and CD8⁺ T cells in a dose-dependent manner. (C) The concentration of proinflammatory T cell cytokines IFN-γ⁺ and IL-17 was measured in the supernatant of the cultures of activated autologous T cells with matching M-MDSC by ELISA. The production of IFN-γ⁺ and IL-17 was significantly reduced in the cultures of T cells with matching M-MDSC at 1:1 ratio in T1D patients (n = 15). No significant suppression of T cell proinflammatory cytokines appeared in HD (n = 3). *p≤0.05, **p≤0.01, ***p≤0.001 (paired t-test).

## M-MDSC suppression of T cell response requires cell-contact and is dependent on TGF-β

The suppressive properties of M-MDSC are generally attributed to arginase-1 production, induction of iNOS and ROS production, TGF-β, IL-10 and many others [8, 47–49]. Neverthe-less, few data have mentioned the mechanism of action in MDSC-mediated T cell suppression in MDSC directly isolated from T1D patients. In our study, we analyzed for the first time the detailed mechanism by which native M-MDSC sorted from peripheral blood of T1D patients mediate T cell suppression.

First, we explored whether M-MDSC-mediated suppression of T cell proliferation also requires cell-cell contact. For this purpose, we used a two-chamber transwell system where autologous CD3/CD28 activated T cells were added to the lower compartment of a plate con-taining Tranwell insert and sorted M-MDSC were added to the upper compartment in an indi-cated rations (1:1, 1:2 and 1:4) and the T cell proliferation was determined by flow cytometry as CFSE dilution. Using this assay, we observed a clear cell contact dependency in M-MDSC-mediated suppression of CD4$^+$ in the 1:1 ratio (Fig 4A). These data demonstrate that direct cell contact between T cells and M-MDSC is crucial for M-MDSC to fulfill their suppressive potential.

Next, we investigated whether the production of iNOS, TGF-β, and arginase-1 by M-MDSC are indispensable for their suppressive function in T1D patients. As shown in Fig 4B, CD4$^+$ T cells were co-cultured with autologous sorted M-MDSC and activated by anti CD3/CD28 beads. The particular inhibitors, L-NMMA–iNOS inhibitor, anti-TGF-β mAb, and Nor-NOHA–arginase -1 inhibitor, were added to the culture in determined dosage. The subse-quent proliferation of T cells was analyzed by flow cytometry as a CFSE dilution. The suppres-sion of T cell proliferation by M-MDSC was significantly dependent on TGF-β and appeared to be independent of iNOS and arginase-1.

## The frequency of M-MDSC negatively correlates with the CD3ζ chain expression on T cells in T1D patients and their at-risk relatives

Another mode of action that M-MDSC use to suppress T cell proliferation is lowering CD3ζ expression on T cells. We analyzed the CD3ζ expression on T cells isolated from PMBC of T1D patients, at-risk relatives, and HD by flow cytometry. The CD3ζ chain expression on T cells was significantly lower in T1D patients in comparison to their at-risk relatives and healthy donors (Fig 5A). There was a clear negative correlation between the percentage of M-MDSC from PBMC and CD3ζ expression on CD4$^+$ cells in T1D patients, at-risk relatives and HD. The statistically significant negative correlation between M-MDSC and CD3ζ expression on CD8$^+$ T cells was observed only in T1D, but not in at-risk relatives nor in HD, despite the simi-lar trend (Fig 5B and 5C).

## M-MDSC from T1D patients are T cell suppressors but less potent than M-MDSC isolated from patients with lung cancer

In cancer patients, MDSC suppress antigen-specific immunity via interaction with regulatory T cells (Tregs). MDSC utilize heterogeneous mechanism to enhance T cell IL-10 expression and reduce IFN-γ production [26, 50, 51]. However, the role and mechanism of action of MDSC in the autoimmune diseases are less explored and limited data are available about the suppression potential of M-MDSC in T1D [24, 36]. We asked the question whether the M-MDSC from patients with autoimmune T1D are equally potent in T cells inhibition as M-MDSC isolated from patients with a malignity. We co-cultured T cells with sorted CD14$^+$

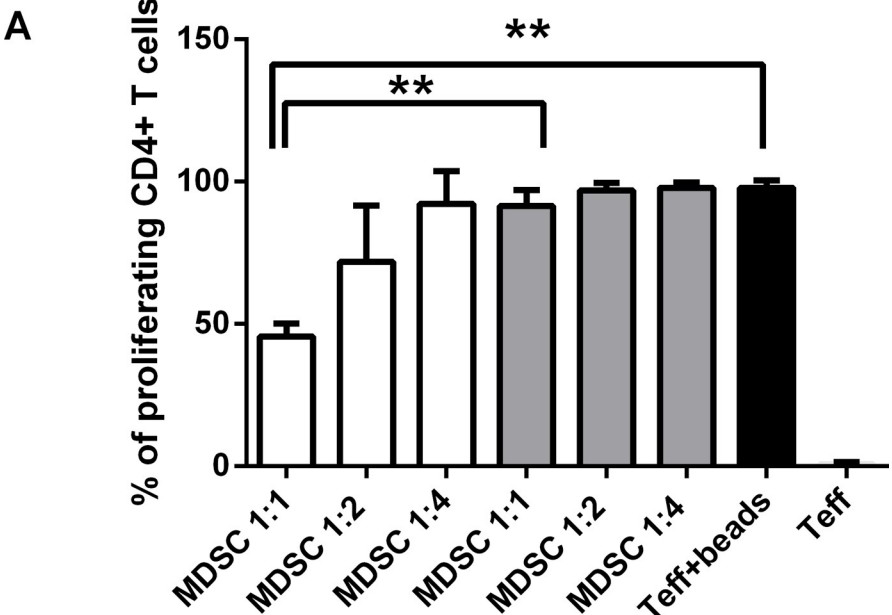

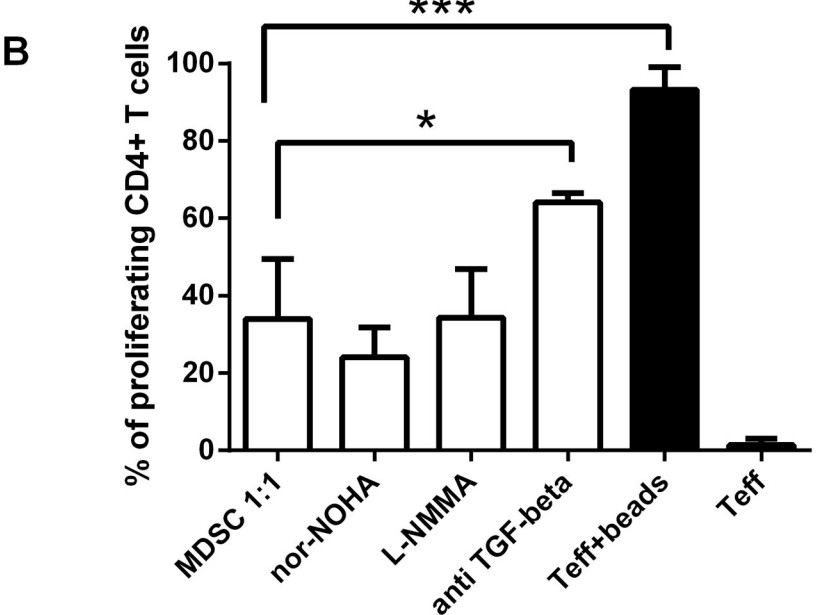

**Fig 4. M-MDSC suppression of T cell response requires cell-contact and is dependent on TGF-β.** (A) The autologous culture of CD4⁺ T cells and sorted M-MDSC in an indicated ratios (T cell: M-MDSC 1:1, 1:2 and 1:4) from PBMC of T1D subjects (n = 4) were performed with the two-chamber transwell system (gray bars) and without the transwell system (white bars). The suppression of CD4⁺ T cell proliferation was significant only in the culture without the transwell system and in the ratio 1:1. The black bars indicate the control proliferation of T effectors (eff) activated by anti-CD3/CD28 beads and without the activation. *p≤0.05, **p≤0.01, ***p≤0.001 (paired t-test). (B) CD4⁺ T cells were co-cultured with autologous sorted M-MDSC from T1D patients (n = 4). The particular inhibitors, Nor-NOHA–arginase-1 inhibitor, L-NMMA–iNOS inhibitor and anti-TGF-β mAb were added. The suppression of T cell proliferation by M-MDSC was significantly dependent on TGF-β and appeared to be independent of iNOS and arginase—1. The black bars indicate the control proliferation of T effectors (eff) activated by anti-CD3/CD28 beads and without the activation. *p≤0.05, **p≤0.01, ***p≤0.001 (paired t-test).

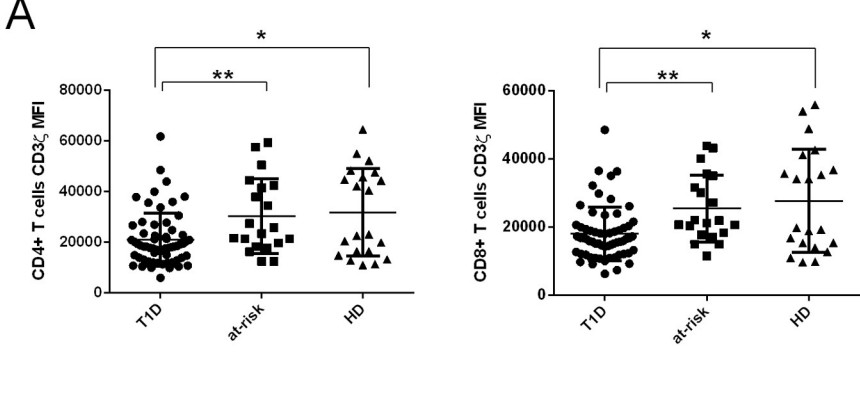

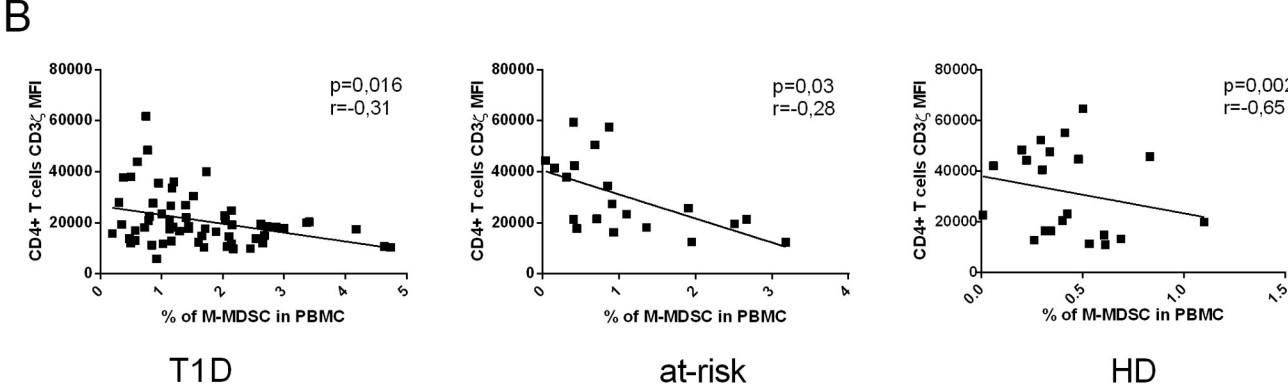

T1D at-risk HD

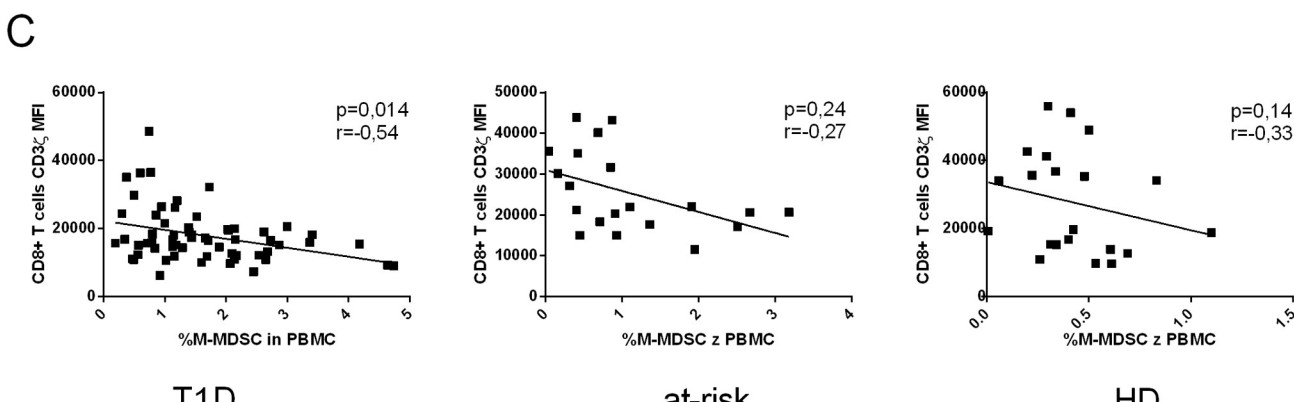

T1D at-risk HD

**Fig 5. Immune suppressive activity of M-MDSC is associated with the down-regulation of CD3ζ expression on T cells.** (A) Down-regulation of CD3ζ on CD4+ and CD8+T cells was observed in T1D patients (n = 63) compared to at-risk relatives (n = 20) and healthy donors (HD) (n = 21). *p≤0.05, **p≤0.01, ***p≤0.001 (Mann-Whitney test). (B, C) The correlation analysis of the frequency of M-MDSC and the level of CD3ζ expression on CD4+ and CD8+ T cells in T1D patients, at-risk relatives and healthy donors (HD). A clear negative correlation was seen between the percentage of M-MDSC and CD3ζ expression on CD4+ T cells in T1D patients, at-risk relatives and HD. The statistically significant negative correlation between

M-MDSC and CD3ζ expression on CD8$^+$ T cells was observed only in T1D, not in at-risk relatives nor in HD, despite the similar trend. Each point represents the value from an individual subject. The r values indicate the correlation index according to Spearman correlation analysis.

HLA-DR $^{low/neg}$ CD33$^{high}$ M-MDSC from either T1D patients, patients with non-small cell lung cancer or HD in an indicated ratio (MDSC/ T cell 1:1, 1:2 and 1:4 ratio). Consistently with our previous findings, we observed the significant suppression of T cell proliferation by M-MDSC in a dose-dependent manner in T1D patients (MDSC/ T cell ratio of 1:1 was the most effective, and the inhibitory function was lost at 1:4 ratio as depicted in Fig 3A). However, the inhibition was weaker than that we observed in MDSC isolated from patients with lung cancer, which suppressed T cell proliferation markedly even at the 1:4 ratio. The suppression effect of M-MDSC isolated from HD was negligible. (Fig 6A and 6B)

## Discussion

In recent years MDSC have been intensively studied for their immunosuppressive potential and several studies have provided a piece of evidence that MDSC might represent a therapeutic strategy for the treatment of T1D. In murine models of autoimmune diabetes, MDSC were able to prevent the destruction of pancreatic islets or suppress diabetogenic T cell function and reverse autoimmune diabetes [35, 52]. However, the protective role of MDSC in human T1D is more controversial and less explored. Whitfield-Larry et al. documented MDSC expansion in the peripheral blood of T1D patients as well as in the peripheral blood and secondary lymphoid organs of NOD mice. Interestingly, MDSC frequency was decreased within the pancreatic islets of mice [36] similarly to the results from Fu et al. showing the inverse correlation between intra-islet MDSC and diabetes progression [53]. These findings support the hypothesis of MDSC involvement in T1D pathogenesis.

In our study, we explored the comprehensive role of M-MDSC in T1D patients and for the first time, we analyzed the molecular mechanism of the suppressive function of M-MDSC directly isolated from the blood of T1D patients. Moreover, we focused on the role of M-MDSC in islet-cells antibody- positive first-degree relatives of T1D patients.

In agreement with the study by Whitfield-Larry [36], we observed the increased frequency of M-MDSC in the peripheral blood of T1D patients compared to healthy donors. In addition to the previous study, we observed significantly elevated levels of M-MDSC in the peripheral blood of at-risk relatives as well. Interestingly, the patients' level of HbA1c may be related to the number of M-MDSC in PBMC. Moreover, we demonstrated a positive correlation between the frequency of M-MDSC and Th17 cells and IFN-γ producing T cells in T1D patients and their at-risk relatives.

The increased frequency of M-MDSC in T1D patients and their relatives could be the consequence of the expansion of various factors that are elevated in chronic inflammatory processes such as autoimmune diseases. Indeed, proinflammatory cytokines such as IL-1, IL-6, and IFN-γ were described to lead to the expansion of MDSC in cancer [7, 8] and these cytokines are also elevated in the serum of diabetic patients as well as in healthy at-risk relatives without the disease onset [54–58]. The hypothesis of the cytokine-mediated accumulation of MDSC supports other studies reporting that MDSC were expanded in the peripheral blood of subjects with other autoimmune conditions. Besides the elevation of MDSC in T1D as mentioned above [36], a noticeable increase in MDSC frequency was described in patients with RA [32], IBD [29, 59] and psoriasis [60, 61].

Next, we divided the T1D patients into two groups based on the level of HbA1c and we observed that the frequency of MDSC was increased in the group with a poor long-term metabolic status (HbA1c level of > 7.5%). The long term hyperglycemia in these patients is strongly

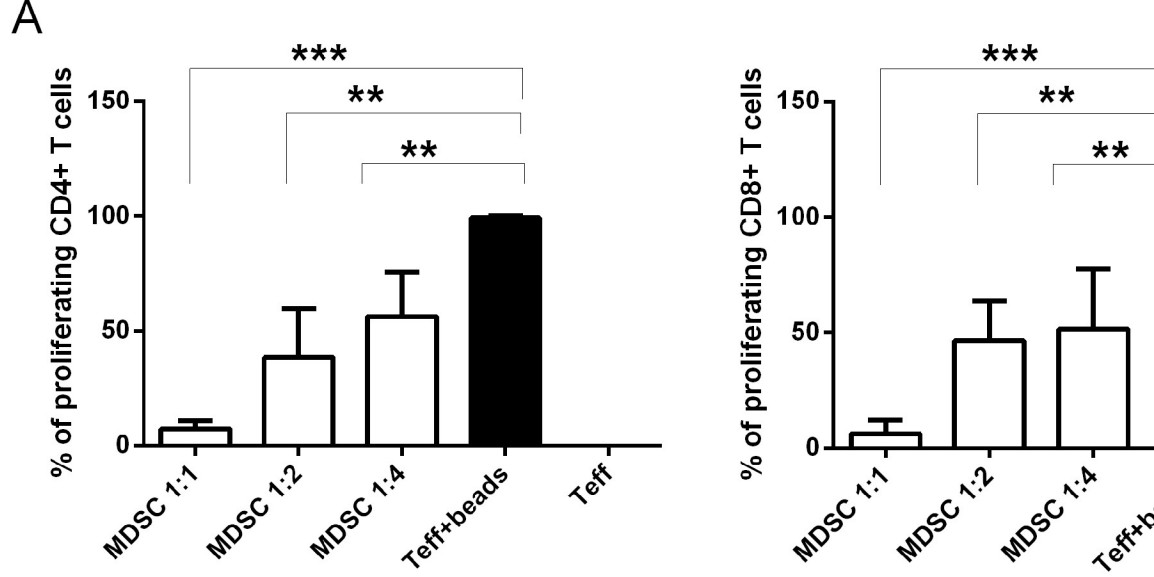

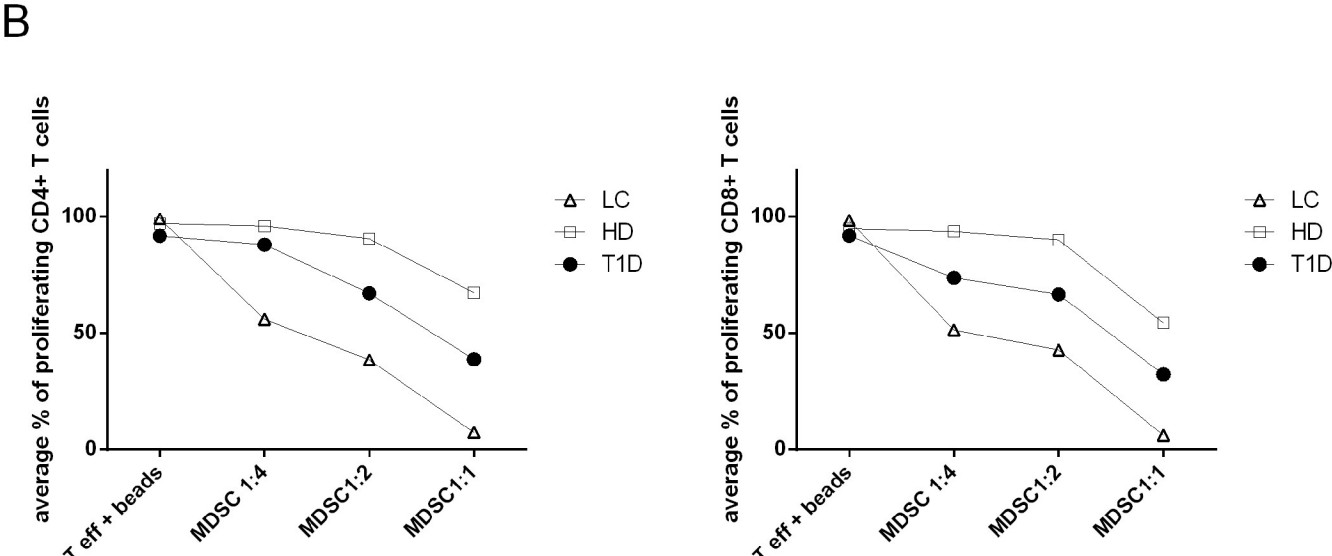

**Fig 6. M-MDSC from patients with lung cancer most suppress T cell proliferation.** (A) M-MDSC sorted from PBMC of the patient with lung cancer (n = 4) were cultured with autologous CD4[+] T cells and CD8[+] T cells. M-MDSC significantly inhibited T cell proliferation in a dose-dependent manner, the T cell/MDSC ratio of 1:1 was the most effective. $^{*}$p≤0.05, $^{**}$p≤0.01, $^{***}$p≤0.001 (paired t-test). (B) Comparison of M-MDSC potential to suppress T cell proliferation in T1D patients (•), lung cancer patients, LC (Δ) and healthy donors, HD (□). The culture of M-MDSC and CD4[+] / CD8[+] T cells from PBMC was performed in an indicated ratio (Tcell/M-MDSC 1:1, 1:2, 1:4) and the percentage of proliferating cells was compered among themselves. The most suppressive potential was observed in M-MDSC isolated from patients with lung cancer.

associated with chronic inflammation milieu and the formation of advanced-glycation end products (AGEs) and their RAGE receptors [40]. RAGE–related expansion of MDSC has been already described in patients with cancer by signalization via members of the S100 protein family [62–64]. Thus we suggest that the AGEs-RAGE interaction in MDSC could contribute to their elevation in patients with poor long-term disease control.

Interestingly, recently published data reveal the role of TNF-α in MDSC expansion and peripheral accumulation [65]. Moreover, TNF-α was also shown to arrest the differentiation of immature MDSC and augment their suppressive properties [66]. Although we did not measure the TNF-α level in the serum of our subjects, a large meta-analysis confirms the significant elevation of TNF-α in the serum of T1D patients [67]. Probably the MDSC expansion in diabetic patients, observed in our study, could be also related to TNF-α elevation.

Further, we demonstrated for the first time a positive correlation between M-MDSC and proinflammatory Th17 cells and IFN-γ producing cells in T1D patients and their at-risk relatives. Th17 cells and IFN-γ producing cells are often elevated in autoimmune disease [1, 45, 54, 68], nevertheless their relation to MDSC varies through the different studies from the positive to negative correlation [32, 34, 69–72]. In rheumatic arthritis MDSC were documented to be simultaneously expanded together with Th17 cells and associated with the disease progression [34]. Contrary, the adoptive transfer of MDSC prevented the autoimmune arthritis in mouse models of RA through inhibiting Th17 cells [69]. This discrepancy is not fully elucidated. The distinctive local environment may influent the various level of MDSC activation as well as the expansion of different MDSC subpopulations [73]. Remarkably, specific *in vitro* studies proved that both human and murine MDSC could promote the Th17 development in IL-1β dependent manner alongside the favorable cytokine milieu, especially IL-6 and TGF-β secreted by MDSC [34, 74] and also exogenous NO [71]. Thus, MDSC which could be elevated in T1D patients due to proinflammatory cytokines perhaps promote concomitant Th17 development via secretion of IL-1 β, IL-6, and TGF-β and/or exogenous NO [71, 75]. Th17 expansion is associated with IL-17 production and IL-17 was described to further induce and activated MDSC with their suppressive properties as self-regulation [76, 77].

The role of INF-γ in elevation of MDSC levels is very comprehensive. Studies in tumor-bearing mice provided a more detailed insight into the interaction between MDSC and IFN-γ producing cells and documented that IFN-γ itself is important for MDSC activation [78]. Similar results were demonstrated in the immune-mediated liver injury in mice where IFN-γ also induces MDSC [79].

Regarding suppressive properties of MDSC, we proved in our *in vitro* experiments that M-MDSC directly isolated from T1D patients keep the immune inhibition properties. MDSC effectively suppressed CD3+ and CD4+ T cells proliferation and the production of IL-17 and IFN-γ cytokines, however, only at high MDSC: T cell ratio. Thus, taken into account our observation that increased MDSC levels were not associated with the decrease of proinflammatory T cell population *in vivo* in T1D patients and their at-risk relatives, we might postulate that MDSC become effective suppressors of T cell only at sufficient ratio to T cells. Although we proved in the proliferation assay that MDSC isolated directly from T1D patients exhibit the suppressive potential *in vitro*, whether MDSC keep the suppression properties also *in vivo* remains unknown. The M-MDSC frequency in the peripheral blood may be too low to fulfill their suppressive potential or they could not be fully suppressive due to the local microenvironment influencing the mechanism of action of MDSC.

We did not see the MDSC expansion in HD which may be caused by generally non proinflammatory milieu in healthy individuals and also the MDSC are probably less activated while they were less potent to suppress the T cell proliferation *in vitro*.

Till now, no data concerning the mechanism of suppression of M-MDSC isolated directly from T1D patients have been published. Thus we explored it in more detail. We observed T-cell receptor CD3ζ chain downregulation on T cells and we attributed this fact to the increased M-MDSC frequency. Although M-MDSC mediated downregulation of CD3ζ chain was generally attributed to arginase—1 [18, 80, 81], in our experiments arginase—1 did not play the role in M-MDSC mediated T cell suppression. Currently, Bian et al. showed that arginase—1 might be neither constitutively expressed in MDSC nor required for MDSC-induced downregulation of CD3ζ chain and T cell proliferation. Moreover other proinflammatory cytokines such as IL-17 and IFN-γ could negatively regulate the arginase—1 expression in MDSC [82]. Thus we suggest that M-MDSC in the group of T1D subjects engage a different mechanism to suppress T cells, preferentially TGF-β production and cell-cell contact between M-MDSC and T cells. It is also worth mentioning that TNF-α, elevated in T1D patients [67], was described to augment the M-MDSC suppressive activity accompanied by T cell CD3ζ chain downregulation [66, 83]. Moreover, MDSC independent effect of lowering CD3ζ chain should be also taken into account. In another immunopathological condition such as RA or SLE, CD3ζ chain was shown to be downregulated via TNF-α itself leading to the proteasomal degradation [84, 85].

When we compared the suppressive function of M-MDSC isolated from T1D and from patients with tumors, the tumor's M-MDSC exhibited even more potent suppressive potential. Probably the cytokines in tumor microenvironment more augment the suppressive effect of M-MDSC whereas M-MDSC in healthy individuals may not be maximally activated/ effective to suppress T cells. Conclude that evidence we postulate that the cytokine milieu regulates the executive molecular mechanism of suppression of M-MDSC and in T1D patients M-MDSC suppressive effect relies on cell-cell contact and TGF-β and is independent of arginase—1.

In conclusion, the inconsistent role of MDSC in autoimmune diseases is more than obvious. Such a discrepancy in the results could be due to the heterogeneity of MDSC, the different inflammatory context in which MDSC interact with T cells and last but not least due to different stages of particular diseases. Even though, our data reported that M-MDSC could play an important disease modulation role in T1D and exhibit a suppressive effect on T cells *in vitro*. The question remains whether M-MDSC are relevant to immune interventional strategy for therapy of T1D since various studies pointed to the contrary role of MDSC in autoimmune diseases. A very attractive query is also the role of MDSC directly at the site of the inflammation in the pancreas but the inaccessibility of their isolation brings obvious obstacles to explore it. An equally interesting question remains whether an *in vitro* cytokine modulation could enhance the suppressive properties of MDSC, similarly to Whitfield-Larry's study [36]. Perhaps another comprehensive study focusing on the suppressive potential and molecular pathways in MDSC helps further elucidate some of the aforementioned questions.

## Supporting information

**S1 Fig. The gating strategy of M-MDSC.**
(PDF)

## Author Contributions

**Conceptualization:** Klára Dáňová, Irena Adkins, Radek Špíšek, Lenka Palová-Jelínková.

**Data curation:** Klára Dáňová, Zdeněk Šumník, Lenka Petruželková.

**Formal analysis:** Anna Grohová.

**Funding acquisition:** Zdeněk Šumník.

**Investigation:** Anna Grohová, Lenka Palová-Jelínková.

**Methodology:** Irena Adkins, Lenka Palová-Jelínková.

**Supervision:** Zdeněk Šumník, Radek Špíšek, Lenka Palová-Jelínková.

**Validation:** Klára Dáňová, Irena Adkins, Lenka Petruželková, Barbora Obermannová, Stanislava Koloušková.

**Writing – original draft:** Anna Grohová.

**Writing – review & editing:** Lenka Palová-Jelínková.

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
