## [Decision Letter · Decision Letter 0]

4 Sep 2020

PONE-D-20-21838

Myeloid - derived supressor cells in Type 1 diabetes are an expanded population exhibiting diverse T-cell suppressor mechanisms

PLOS ONE

Dear Dr. Grohova

Thank you for submitting your manuscript to PLOS ONE. After careful consideration, we feel that it has merit but does not fully meet PLOS ONE’s publication criteria as it currently stands. Therefore, we invite you to submit a revised version of the manuscript that addresses the points raised during the review process.

The experiments are well designed though analysis and explanation of data lacks clarity. The main message of the study can be discussed with changes suggested by the two reviewers.

We look forward to receiving your revised manuscript.

Kind regards,

Veena Taneja

Academic Editor

PLOS ONE

Additional Editor Comments:

The message of the article does not come across clearly mainly due to certain experimental details that need clarification. Ambiguity in certain discussion needs to be revised.

Journal Requirements:

2. Thank you for including your ethics statement:  "All subjects provided written consent following the local Ethic Committee (University Hospital in Moto, Prague, Czech Republic).".   

Please amend your current ethics statement to confirm that your named institutional review board or ethics committee specifically approved this study.

We note that one or more of the authors are employed by a commercial company: SOTIO a.s..

3.1.Please provide an amended Funding Statement declaring this commercial affiliation, as well as a statement regarding the Role of Funders in your study. If the funding organization did not play a role in the study design, data collection and analysis, decision to publish, or preparation of the manuscript and only provided financial support in the form of authors' salaries and/or research materials, please review your statements relating to the author contributions, and ensure you have specifically and accurately indicated the role(s) that these authors had in your study. You can update author roles in the Author Contributions section of the online submission form.

3.2. Please also provide an updated Competing Interests Statement declaring this commercial affiliation along with any other relevant declarations relating to employment, consultancy, patents, products in development, or marketed products, etc.  

We note that you have included the phrase “data not shown” in your manuscript. Unfortunately, this does not meet our data sharing requirements. PLOS does not permit references to inaccessible data. We require that authors provide all relevant data within the paper, Supporting Information files, or in an acceptable, public repository. Please add a citation to support this phrase or upload the data that corresponds with these findings to a stable repository (such as Figshare or Dryad) and provide and URLs, DOIs, or accession numbers that may be used to access these data. Or, if the data are not a core part of the research being presented in your study, we ask that you remove the phrase that refers to these data.

Reviewers' comments:

Reviewer's Responses to Questions

**Comments to the Author**

1. Is the manuscript technically sound, and do the data support the conclusions?

Reviewer #1: Partly

Reviewer #2: Partly

2. Has the statistical analysis been performed appropriately and rigorously? 

Reviewer #1: Yes

Reviewer #2: No

3. Have the authors made all data underlying the findings in their manuscript fully available?

Reviewer #1: Yes

Reviewer #2: Yes

4. Is the manuscript presented in an intelligible fashion and written in standard English?

Reviewer #1: Yes

Reviewer #2: Yes

5. Review Comments to the Author

Reviewer #1: Summary: The authors are addressing the role of monocytic myeloid derived suppressor cells (M-MDSC) in the pathogenesis of T1D, mainly because there has been a lot of conflicting reports as their role in the pathogenesis of diseases. The authors conducted well designed experiments to determine this, such as evaluating the levels of M-MDSCs and any associations/correlations with different markers of T1D in patients and compared them to healthy controls. They also conducted in vitro suppression assays to determine that the circulating M-MDSCs from T1D patients are potent suppressors of T cells, albeit not as potent as the M-MDSCs from patients with squamous lung carcinoma. Their results are consistent with results from previous studies in that they found what appear to be inconsistent functions of the M-MDSCs in the pathogenesis of T1D, meaning elevated numbers of circulating M-MDSCs in T1D patients as compared to healthy controls, while at the same time finding that the M-MDSCs are capable of suppressing T cell function in vitro. My main concern with the manuscript is that the general message of the manuscript drifts from implying that M-MDSCs could play a regulatory role in T1D and be given to treat patients in the introdcution to the first line of the discussion that states, “the inconsistent role of MDSC in autoimmune diseases is more than obvious”. I believe that this manuscript should be published because the experiments are well designed, and the data acquired supports that first line in the discussion; however, I believe that the introduction needs to be modified to reflect that the inconsistent role of M-MDSC has been seen with other diseases. This would put the reader into the frame of mind that the ensuing results will be consistent with this observation from other publications. Also too, a sentence in the discussion should be added that states that caution should be taken when considering therapies with M-MDSCs, since their function in different settings/diseases can be completely different, potentially even, unexpected.

Major Criticisms:

1) From the experimental design, it is not clear as to which T cells the MDSCs are suppressing in the suppression assay. Please add some clarification as to the possibility of regulatory T cells being suppressed as well in this assay.

2) Was there a correlation between M-MDSC and HbA1c levels? Please provide another panel in figure 1 showing a correlation analysis and M-MDSC levels and HbA1c levels.

3) Please clarify with text in the results section, in the transition area between Line 301 and 307. On line 301, it states that the inhibition assay demonstrated that M-MDSC from the T1D patients exerted their suppression of T cells in an arginase1 independent fashion, but line 306 and line 307 introduce the result of decreased CD3e chain by saying that decreased CD3e is an arginase1 dependent event.

4). The discussion is the best (and least confusing) section of text in the whole manuscript. Mainly because it clearly states in the first line, “ the inconsistent role of MDSC in autoimmune diseases is more than obvious” Lines 48-51 in the introduction should be expanded to make this point early on with previous publications. Intent being to set the stage for the rest of the manuscript that MDSCs will act very differently in different pathological settings.

5). Based on the result that a 1:1 ratio is necessary for suppression in the in vitro suppression assays, what is the expected ratio of M-MDSC and effector T cells in the periphery of the T1D patients? Meaning, is the ratio too low for M-MDSC to be effective in the T1D patients? There is a discrepancy between a correlation of high levels of inflammatory T cells and high levels of M-MDSC with the fact that the M-MDSCs from T1D patients can suppress inflammatory T cells in the in vitro suppression assay.

6). Line 248: Why is it expected that the M-MDSCs of healthy should not be suppressive? Isn’t this another example of the odd functions that M-MDSCs exhibit in different settings?

Minor Criticisms:

1). What is the meaning of casual on line 47?

2). Line 53: Typo? MDSC?

3). Line 90: Please clarify the connection between MDSC and glycated hemoglobin, meaning as a marker of disease severity.

4). The scale on the Y axis for panels in 2B and 2C should all be the same.

Reviewer #2: The authors show a well crafted series of experiments in biomedical/immunological aspects of MDSCs and t cell function in T1D patients and their relationship to high risk groups. An interesting link is presented in the role of MDSCs and t cell function in at risk groups as well as in T1D patients and several questions and comments arose during the revision of the manuscript:

1) Further details should be included in the selection criteria of the groups.

2) In the clinical data it should also include the anthropometric data, laboratory analysis data, ethnicity, HOMA-IR, insulin levels, Hb1ac%, auto-antibody levels, etc. This, given that several of these variables are known to influence MDSC frequency.

3) No flow cytometry data is provided in the form of a gating strategy and showing the differences of MDSC populations when comparing at least a few cases of each group. Also, it is advised that the FCS files be available in flow cytometry repositories.

4) In the statistical analysis, it is mentioned that paired/unpaired experiments were performed but it should be indicated in the figure legends for each case. Also, no definition of normality and therefore no nonparametric tests were used in case of non-normal data. A correction for partial correlations should be used in the case that variables as those proposed to be included in table 1 show statistically significant differences among groups.

5) In some cases it is clear that non-parametric correlation should´ve been performed. This needs to be corrected for all correlations and correct the conclusiones derived from those analysis.

6) Further clarification should be done for the MDSC-Th17 relationship both in the discussion and in the results sections as it is difficult to follow the rationale and the conclusions derived from this part.

7) In the discussion section, no explanation regarding MDSCs elevated levels and increased function are discussed in the context of the proliferation assays.

6. PLOS authors have the option to publish the peer review history of their article (what does this mean?). If published, this will include your full peer review and any attached files.

Reviewer #1: No

Reviewer #2: No

---

## [Author Response · Author response to Decision Letter 0]

18 Oct 2020

Dear Prof. Taneja, 

Thank you for your decision on our manuscript " Myeloid-derived suppressor cells in Type 1 diabetes are an expanded population exhibiting diverse T-cell suppressor mechanisms". 

We are sending the rebuttal letter explaining each of the reviewers´ suggestions and comments. We found the comments very helpful and constructive. We revised the reviewers´ comments carefully point-by-point, we accepted and tried to meet all the recommendations. We believe that the objections raised by the reviewers can be addressed and implemented in the revised manuscript. We are confident that the revised manuscript would be easier to understand and has a more fluent scientific discourse.

Ms. Ref. No.: PONE-D-20-21838

Title: Myeloid-derived suppressor cells in Type 1 diabetes are an expanded population exhibiting diverse T-cell suppressor mechanisms

Comments to reviewer 1:

Q1: From the experimental design, it is not clear as to which T cells the MDSCs are suppressing in the suppression assay. Please add some clarification as to the possibility of regulatory T cells being suppressed as well in this assay.

A1: In our experiments, we decided to choose the whole population of CD3+ T cells, purified by appropriate kit, and then we looked separately at the proliferation of CD4+ T cells and CD8+ T cells as the main T cell population. We did not assess the T regulatory cells population although naturally occurring T regulatory cells are presented in the CD4+ population yet in a minute amount.

Q2: Was there a correlation between M-MDSC and HbA1c levels? Please provide another panel in figure 1 showing a correlation analysis and M-MDSC levels and HbA1c levels.

A2: We did not see any correlation between the HbA1c level and the frequency of M-MDSC in PBMC as amended in Fig 1. Despite no linear correlation, the patients with worse long-term metabolic conditions seem to have a higher percentage of M-MDSC in their PBMC than those with better metabolic control. Likely the long-term hyperglycemia, reflected by the elevation of HbA1c, may promote the M-MDSC induction, although the reliance is not in a linear correlation with the level of HbA1c.

Q3: Please clarify with text in the results section, in the transition area between Line 301 and 307. On line 301, it states that the inhibition assay demonstrated that M-MDSC from the T1D patients exerted their suppression of T cells in an arginase1 independent fashion, but line 306 and line 307 introduce the result of decreased CD3e chain by saying that decreased CD3e is an arginase1 dependent event.

A3: We checked the current literature describing the role of M-MDSC-produced arginase-1 in the down-regulation of CD3ζ chain on T cells and we found contradictory data.

Indeed, for MDSC inhibiting T cells, the arginase-1 was shown to impede T cell proliferation and depletion of L-arginine is associated with a reduction of the T cell receptor (TCR) subunit CD3ζ, leading to a diminished TCR response (Bronte et al., Nat Rev Immunol 2005, 641-654; Rodriguez et al., The Journal of biological chemistry. 2002;277(24):21123-9.). 

On the other hand, the current study showed that arginase-1 is neither constitutively expressed in nor required for MDSC-mediated inhibition of T cell proliferation (Bian et al., Eur J Immunol 2018, 1046–1058). Furthermore, inhibition of arginase-1 activity by the chemical inhibitor nor-NOHA, has been shown to rescue T cell proliferation as documented in Ochoa et al., Clin Cancer Res 2007: 1627-1634, however, other studies using the same arginase inhibitor failed to rescue T cell proliferation as documented in Yao et al., PlosOne. 2016; 11(2):e0149948.

In summary, these varied or controversial studies point to the lack of clarity surrounding the mechanism(s) that control arginase-1 expression in MDSC, as well as how MDSC inhibit T cell proliferation.

Thus, we decided to remove the statement that the down-regulation of CD3ζ expression on T cells by M-MDSC is generally attributed to arginase -1. Next, we added the text discussing the contradictory role of arginase-1 in M-MDSC and the MDSC-mediated CD3ζ chain lowering to the Discussion (lines 472- 487). Our data suggest that in our subject M-MDSC may utilize different mechanisms apart from arginase-1 to suppress CD3ζ chain expression and T cell proliferation.

Q4: The discussion is the best (and least confusing) section of text in the whole manuscript. Mainly because it clearly states in the first line, “ the inconsistent role of MDSC in autoimmune diseases is more than obvious” Lines 48-51 in the introduction should be expanded to make this point early on with previous publications. Intent being to set the stage for the rest of the manuscript that MDSCs will act very differently in different pathological settings.

A4: The inconsistent role of MDSC in the autoimmune condition is further discussed in the Introduction section. We point out this message in lines 50-51 and 71 -86.

Q5: Based on the result that a 1:1 ratio is necessary for suppression in the in vitro suppression assays, what is the expected ratio of M-MDSC and effector T cells in the periphery of the T1D patients? Meaning, is the ratio too low for M-MDSC to be effective in T1D patients? There is a discrepancy between a correlation of high levels of inflammatory T cells and high levels of M-MDSC with the fact that the M-MDSCs from T1D patients can suppress inflammatory T cells in the in vitro suppression assay.

A5: As we mention in the Discussion, Th17 cells and IFN-γ producing cells are often elevated in autoimmune disease, nevertheless their relation to MDSC varies through the different studies from the positive to negative correlation. This discrepancy is not fully clarified. Based on the current literature we supposed that MDSC and Th17 cells interplay could promote reciprocal induction. MDSC were described to induce Th17 cells via IL-1�, IL-6, TGF-� and exogenous NO. Next, Th17 cells produce IL-17 cytokine leading to further MDSC expansion. Also IFN-� could promote the differentiation of MDSC. Nevertheless based on our experiments we concluded that MDSC become effective in T cell suppression only at a sufficient amount, as in our in vitro experiments MDSC suppress T cell proliferation only in the 1:1 ration (MDSC/ T cells). We expected that the frequency of M-MDSC in the peripheral blood of T1D patients is much lower than the frequency of effector T cells. Generally, about 2% as also reviewed in current literature. The ratio could be too low to suppress the effector T cells expanded in T1D patients. 

Also, M-MDSC in T1D patients may not be fully suppressive. This could be due to various cytokine milieu in vivo and as we pointed out in the Conclusion perhaps in vitro cytokine modulation could enhance the suppressive properties of MDSC, similarly to Whitfield-Larry’s study.

We tried to clarify this problem in the Discussion section lines 435 -451 and 460-468.

Q6. Line 248: Why is it expected that the M-MDSCs of healthy should not be suppressive? Isn’t this another example of the odd functions that M-MDSCs exhibit in different settings?

A6: It seems that MDSC exhibit different suppressive ability under steady-state conditions compered to different pathological conditions such as infection, autoimmune diseases, or cancer. Re-programming of myeloid cells toward fully suppressive MDSC can be considered the result of a multistep process orchestrated by various factors produced during acute and chronic inflammation or by the tumor microenvironment. Those factors are not present under steady-state conditions in healthy donors and might explain the weak suppressive function of MDSC in HD.

Our data demonstrating a weaker suppressive effect of M-MDSC isolated from HD compared to those from T1D patients or cancer patients are in agreement with data published by Cao et al, in J of Investigative Dermatology in 2016: 1801-1810 demonstrating weak inhibition of IFN-� produced by autologous T cells after co-culturing with MDSC isolated from HD in comparison to MDSC isolated from patients with psoriasis or melanoma patients. Next, Li et al demonstrated in Oncotarget 2017:24380–24388 weaker suppressive ability of MDSC from HD compared to MDSC from patients with hepatocellular carcinoma. 

We clarify the problem in discussion lines 488-494.

Minor comments:

Q1). What is the meaning of casual on line 47? 

A1). Meaning of casual is the specific treatment. No symptomatic.

Q2). Line 53: Typo? MDSC? 

A2). Corrected 

Q3). Line 90: Please clarify the connection between MDSC and glycated hemoglobin, meaning as a marker of disease severity. 

A3). Clarified in the text. (HbA1c as a marker of the metabolic condition of T1D patients reflecting the long term glycemia level).

Q4). The scale on the Y axis for panels in 2B and 2C should all be the same.

A4). Corrected in Figures.

Comments to reviewer 2:

Q1: Further details should be included in the selection criteria of the groups.

A1: For our trial we choose the pediatric patients with T1D, both with recent onset and with long-term T1D duration, both after the establishment of normoglycemia. The first degree relatives were subjects up to the age of 18 years whose at least one sibling suffer from T1D manifested up to the age of 20 years. These subjects fulfilled the criteria for the risk of T1D based on the genotyping and testing for islet-specific autoantibodies (GAD-AB, ANTI- IAA and IA2-AB). We added the selection criteria in the Method section.

 Q2: In the clinical data it should also include the anthropometric data, laboratory analysis data, ethnicity, HOMA-IR, insulin levels, Hb1ac%, auto-antibody levels, etc. This, given that several of these variables are known to influence MDSC frequency.

A2: We amended the Table 2 in the Method section with supplementary data available in our subjects.

Q3: No flow cytometry data is provided in the form of a gating strategy and showing the differences of MDSC populations when comparing at least a few cases of each group. Also, it is advised that the FCS files be available in flow cytometry repositories.

A3: The gating strategy was depicted in the separate Figure and amended as a Supplementary Figure 1.

Q4: In the statistical analysis, it is mentioned that paired/unpaired experiments were performed but it should be indicated in the figure legends for each case. Also, no definition of normality and therefore no nonparametric tests were used in case of non-normal data. A correction for partial correlations should be used in the case that variables as those proposed to be included in table 1 show statistically significant differences among groups.

A4: The data were tested again with Shapiro-Wilk normality test and then the statistical analyses were performed with Mann-Whitney test when needed. We corrected the results in the text and the Figures.

Q5: In some cases it is clear that non-parametric correlation should´ve been performed. This needs to be corrected for all correlations and correct the conclusions derived from those analysis.

A5: We reassessed the correlation analysis and used the Spearman correlation analysis for non-parametric data. We corrected the results in the text and in Figures.

Q6: Further clarification should be done for the MDSC-Th17 relationship both in the discussion and in the results sections as it is difficult to follow the rationale and the conclusions derived from this part.

A6: Th17 cells are often elevated in autoimmune diseases; nevertheless, their relation to MDSC varies through the different studies from the positive to negative correlation. This discrepancy is not yet fully clarified. Based on the current literature we supposed that MDSC and Th17 cells interplay could promote reciprocal induction. MDSC were described to induce Th17 cells via IL-1�, IL-6, TGF-� and exogenous NO. Next, Th17 cells produce IL-17 cytokine leading to further MDSC expansion. Thus, in our subjects, we observed a concomitant expansion of MDSC and Th17 cells. Next, MDSC in T1D patients might be elevated due to other pro-inflammatory cytokines such as IL-1, IL-6 and IFN-�.

We discussed the problem in the text in lines 239 and 244 in the result section and in the discussion, lines 435-451.

Q7: In the discussion section, no explanation regarding MDSCs elevated levels and increased function are discussed in the context of the proliferation assays.

A7: In the proliferation assay we observed that M-MDSC isolated from PBMC in T1D patients were able to suppress T cell proliferation. But the suppressive effect of MDSC was dose-dependent (1:1 MDSC/Tcells). Why this finding does not reflect the situation in vivo and we described a positive correlation between the number of M-MDSC and Th17 cells in PBMC in T1D patietns.is not fully elucidated. We supposed that M-MDSC in T1D patients are not in the high ratio to T cell effectors to fulfill their suppressive potential. There could be a threshold and MDSC become effective suppressors of T cell only in sufficient amount.

Moreover, the MDSC in vivo could be modulated by the local microenvironment and not be fully suppressive and activated or there could be another negative regulation for MDSC which was absent in vivo. 

We discuss this finding in the lines 460-468.

We thank the reviewers for worthy comments and interesting points and tried to rework the study more comprehensible. We hope the all the changes we made will meet your approval.

Looking forward to your reply,

Yours sincerely

Anna Grohová, M.D.

Department of Immunology

Charles University, 2nd Medical School

V Úvalu 84, Prague 5

Czech Republic

e-mail: anna.grohova@gmail.com

---

## [Decision Letter · Decision Letter 1]

27 Oct 2020

Myeloid - derived supressor cells in Type 1 diabetes are an expanded population exhibiting diverse T-cell suppressor mechanisms

PONE-D-20-21838R1

Dear Dr. Grohova,

We’re pleased to inform you that your manuscript has been judged scientifically suitable for publication and will be formally accepted for publication once it meets all outstanding technical requirements.

Kind regards,

Veena Taneja

Academic Editor

PLOS ONE

Additional Editor Comments (optional):

Reviewers' comments:

Reviewer's Responses to Questions

**Comments to the Author**

1. If the authors have adequately addressed your comments raised in a previous round of review and you feel that this manuscript is now acceptable for publication, you may indicate that here to bypass the “Comments to the Author” section, enter your conflict of interest statement in the “Confidential to Editor” section, and submit your "Accept" recommendation.

Reviewer #1: All comments have been addressed

Reviewer #2: All comments have been addressed

2. Is the manuscript technically sound, and do the data support the conclusions?

Reviewer #1: Yes

Reviewer #2: Yes

3. Has the statistical analysis been performed appropriately and rigorously? 

Reviewer #1: Yes

Reviewer #2: Yes

4. Have the authors made all data underlying the findings in their manuscript fully available?

Reviewer #1: Yes

Reviewer #2: No

5. Is the manuscript presented in an intelligible fashion and written in standard English?

Reviewer #1: Yes

Reviewer #2: Yes

6. Review Comments to the Author

Reviewer #1: (No Response)

Reviewer #2: The recommendations and corrections were correctly addressed. The authors have significantly improved the methods section as well as the correct definition of statistical tests. Also, the figures for the definition of the intended cell populations of interest were included. The cell populations being regulated by MDSC´s and the molecular mechanisms may be the focus of future work.

7. PLOS authors have the option to publish the peer review history of their article (what does this mean?). If published, this will include your full peer review and any attached files.

Reviewer #1: No

Reviewer #2: No

---

## [Editor Report · Acceptance letter]

9 Nov 2020

PONE-D-20-21838R1 

Myeloid - derived suppressor cells in Type 1 diabetes are an expanded population exhibiting diverse T-cell suppressor mechanisms 

Dear Dr. Grohová:

I'm pleased to inform you that your manuscript has been deemed suitable for publication in PLOS ONE. Congratulations! Your manuscript is now with our production department. 

Kind regards, 

on behalf of

Dr. Veena Taneja 

Academic Editor

PLOS ONE